# A Design Method of Two-Dimensional Subwavelength Grating Filter Based on Deep Learning Series Feedback Neural Network

**DOI:** 10.3390/s22207758

**Published:** 2022-10-13

**Authors:** Jun-Hua Guo, Ying-Li Zhang, Shuai-Shuai Zhang, Chang-Long Cai, Hai-Feng Liang

**Affiliations:** School of Optoelectronic Engineering, Xi’an Technological University, Xi’an 710021, China

**Keywords:** subwavelength grating sensors, backward design, neural network, deep learning

## Abstract

Subwavelength grating structure has excellent filtering characteristics, and its traditional design method needs a lot of computational costs. This work proposed a design method of two-dimensional subwavelength grating filter based on a series feedback neural network, which can realize forward simulation and backward design. It was programed in Python to study the filtering characteristics of two-dimensional subwavelength grating in the range of 0.4–0.7 µm. The shape, height, period, duty cycle, and waveguide layer height of two-dimensional subwavelength grating were taken into consideration. The dataset, containing 46,080 groups of data, was generated through numerical simulation of rigorous coupled-wave analysis (RCWA). The optimal network was five layers, 128 × 512 × 512 × 128 × 61 nodes, and 64 batch size. The loss function of the series feedback neural network is as low as 0.024. Meanwhile, it solves the problem of non-convergence of the network reverse design due to the non-uniqueness of data. The series feedback neural network can give the geometrical structure parameters of two-dimensional subwavelength grating within 1.12 s, and the correlation between the design results and the theoretical spectrum is greater than 0.65, which belongs to a strong correlation. This study provides a new method for the design of two-dimensional subwavelength grating, which is quicker and more accurate compared with the traditional method.

## 1. Introduction

Subwavelength grating (SWG) sensors have shown attractive application prospects in medical diagnosis [1], environmental monitoring [2], photonic detection [3], and so on, due to its small footprint and compatibility with CMOS manufacturing processes. The filtering quality determines the performance of SWG sensors [4], such as signal-to-noise ratio, sensitivity, etc. Therefore, the design of the filter is the top priority in the development of SWG sensors.

In the past few decades, many researchers have undertaken a detailed study on grating theory, grating analysis, and its application and obtained many achievements. Petit and Marechal et.al presented a rigorous modal theory for grating for the first time in 1960 and put forward two grating calculation methods, such as the integral method and differential method. Many grating theories emerged from the 1970s to the early 1990s, such as the modal method, rigorous coupled wave analysis (RCWA), and Yang-Gu algorithm [5,6,7,8]. In the early 1990s, Professor Li Li-Feng of Tsinghua University improved the stability and convergence of the existing solving algorithm [9,10]. So far, the forward simulation method of subwavelength grating is basically perfected.

Based on the forward simulation method of subwavelength grating, many scholars have carried out research on the design of subwavelength grating. Ye Yan et al. [11] designed two-dimensional subwavelength grating based on the RCWA. The test results showed that the peak transmittances of the red, green, and blue filters are all above 71%. Chen Juan et al. [12] adopted the finite-difference time-domain (FDTD) algorithm to the numerical simulation analysis of rectangular and cylindrical gratings. They designed and researched the two types of two-dimensional subwavelength grating and verified the potential connection between the structural parameters and diffraction characteristics of gratings. Li Caiyu et al. [13] realized the required spectral response by adjusting and optimizing the geometric structure parameters of the grating, and solved the reflectivity of subwavelength grating by combining effective medium theory and transmission matrix. Gao Jian et al. [14] explored the influence of the various structural parameters of the grating on its transmission characteristics through the control variable method and the optimal structure parameters of the designed grating calculated by the RCWA and particle swarm optimization algorithm.

In conclusion, the traditional design method of a two-dimensional subwavelength grating is to preset up the grating geometric structure and simulate its spectral characteristics, then repeatedly adjust the geometric structure parameters of the grating and calculate the optimal spectral response curve by iteration, which is called forward simulation design. In the design, spectral curves are obtained by solving Maxwell’s equations. Which is based on the rigorous coupled wave analysis (RCWA), the finite difference time domain (FDTD), the finite element modeling (FEM), and so on, generally. In terms of a grating structure optimization, traditional numerical optimization methods, such as genetic algorithm [15], fastest descent method, Newton method, and conjugate gradient method [16], have relatively fast convergence speed and high calculation accuracy. However, the result is a local optimal solution. For the global optimization problem, the descending rail method and tunnel method have the advantages such as rapid convergence high computational efficiency. However, their algorithms are complex and the probability of obtaining the global extremum is not high [17]. Meanwhile, the Monte-Carlo random test method and the simulated annealing algorithm are easy to implement, with shortcomings such as slow convergence rate and low computational efficiency [18,19].

For the above algorithms, the resource consumption required for each new grating design is superimposed. Meanwhile, the previously generated data during the optimization process will be discarded, only the optimal results are retained; therefore, the need to consume a lot of computational costs. The reason is that the above optimization algorithms are all essentially based on forward simulation design ideas that depend on the solution of Maxwell’s equations. Whereas the real backward design should reverse simulation, solve, and obtain the geometric parameters of adaptive grating according to the demanded spectral curve. For that, the traditional design method has encountered a bottleneck, and the corresponding design algorithms are rarely reported.

In order to overcome the core problems faced by the traditional design optimization method, and improve the design efficiency of two-dimensional subwavelength grating, a series feedback neural network based on deep learning is built in this paper. In which the forward neural network simulates the spectral response and the backward neural network designs the grating parameters. Through learning and training, the designed series feedback neural network can predict the spectral response of subwavelength grating without solving Maxwell’s equations again, and can accurately reverse design the two-dimensional grating structure. This method realized the real reverse design of two-dimensional subwavelength grating without iterative optimization, and with the optimization design time shortened by several orders of magnitude.

## 2. Design Method

### 2.1. Physical Model of Two-Dimensional Subwavelength Grating

In this paper, a conventional two-dimensional subwavelength grating is taken as the research object, and a design method based on deep learning series feedback neural network is established. The structural model of gratings is shown in Figure 1. Considering the production of subwavelength grating, the grating is mainly composed of an Al grating layer, SiO_2_ base, and two waveguide layers of SiO_2_ and Si_3_N_4_. The geometric structure and shape parameters of the two-dimensional sub-wavelength grating are shown in Table 1.

The effect of grating shape on the diffraction characteristics of the filter is analyzed by taking the structure of a single-layer guide film resonance filter as an example. Rsoft software is used to simulate the designed grating. It is assumed that the refractive index of incident medium is the same as that of the base layer, and the incident light is TE polarized light vertically incident upon the grating surface. The variation of spectral response curves under different grating shapes is shown in Figure 2. The shape of the grating affects the peak value of the spectral response and the position of the peak value changes with the grating shape. Therefore, the effect of the grating shape cannot be ignored. In this paper, five different grating shapes are considered, including circle, triangle, square, pentagon, and hexagon.

### 2.2. Data Set

In this paper, Rsoft software based on rigorous coupled wave analysis and Python Pyautogui module are used to simulate and generate the data set required for the establishment of neural network, respectively, in which the TE mode light is incident on the sample surface perpendicularly. Due to the high degree of freedom of two-dimensional sub-wavelength grating, in order to achieve sufficient data volume, a data set containing 46080 data is established with fully considering the structural parameters of the grating, such as the grating period, grating height, duty cycle, and grating shape. Figure 3 shows the spectral response curves of a part two-dimensional sub-wavelength grating in the data set.

As a discontinuous classification value, the shape features of the two-dimensional sub-wavelength grating cannot be directly introduced into the neural network. Therefore, it is digitized and converted into binary features based on the one-hot encoding. Each shape feature corresponds to an independent register bit, and only one register bit is valid at any time. Thus, it forms a sparse matrix representing the shape feature. The code of the grating shape is shown in Table 2.

### 2.3. Establishment of the Neural Network

#### 2.3.1. Network Structure and Evaluation Criteria

In this paper, a design method of two-dimensional sub-wavelength grating based on deep learning is established, and the neural network is shown in Figure 4. The neural network is a bidirectional model, containing the forward process of simulation represented in the blue arrow and the backward process of the design shown by the red arrow. The grating structure parameters, such as grating height H, SiO_2_ Waveguide layer thickness H_1_, Si_3_N_4_ Waveguide layer thickness H_2_, grating period Λ, duty ratio F, and grating shape are selected as the interaction parameters of the neural network. The forward simulation model is set as the layered double-input mode. The geometric structure parameters of the grating are inputted from the first input layer, which contains five nodes. The grating shape is the second input introduced into the second input layer of the neural network. The forward simulation model takes the spectral response curve as the only output. The input of the backward design model is the quantification of the spectral response, which is expressed as a series of characteristic points for a total of 61. The geometric structure and shape parameters of the grating are synchronously output by the design model.

80% of the data set generated above is randomly selected as training data. Then, it is input into the neural network shown in Figure 4 for training and learning. In order to facilitate the regulation of the training process of the neural network, the mean square error loss function (MSE loss) [20] is introduced as an evaluation index, which is defined as follows for the forward simulation model:(1)$$Loss\;function=MSE(\hat{T},T)=\frac{1}{N}\sum_{i=1}^{N}{\left(\hat{t_{i}}-t_{i}\right)}^{2},$$
where $\hat{T}$ is the spectral curve predicted by the neural network, T is the spectral curve obtained from numerical simulation, $\hat{t_{i}}$ and $t_{i}$ are the values of their characteristic quantization points, respectively, and *N* is the number of output nodes.

For the backward design model, the loss function is:(2)$$Loss\;function=MSE(\hat{Q},Q)=\frac{1}{2}\sum_{i=1}^{N}{\left(\hat{q_{i}}-q_{i}\right)}^{2},$$
where $\hat{Q}$ and Q are the real and neural network designed geometric structure parameters of grating, $\hat{q_{i}}$ and $q_{i}$ are the specific values, such as the grating height, SiO_2_ waveguide layer thickness, Si_3_N_4_ waveguide layer thickness, grating period, duty ratio, and grating shape.

One of the two outputs of the two-dimensional sub-wavelength grating design neural network is the grating shape, which is a classification problem. Therefore, its loss function is different and calculated by the categorical cross-entropy loss [20]:(3)$$Loss\;function=-\sum_{i=1}^{N}y_{i}log\hat{y_{i}},$$

As a classification problem, the data of the grating shape is converted into the one-hot binary features. Therefore, the number set of $y_{i}$ value in the above formula is {0,1}. The loss function of the grating shape focuses on the only result, that is, geometry Softmax label classification.

Figure 5 shows the training results of the neural network. It can be seen from Figure 5a that the forward simulation model has a fast convergence speed and low value of loss function, which fell to just over 0.032 after 1000 iterations. However, the loss function of the backward design model is very higher. As Figure 5b shows, after 1000 iterations, the value of the loss function remained 0.506. This suggests that the iteration of the backward design model cannot provide converge. The problem of non-convergence is mainly caused by the non-uniqueness of data. A small batch size and a large number of training samples must be set as possible for the optimization degree of the neural network. The large amount of data inevitably produces similar spectral response curves. When a similar spectral response corresponds to different grating structure parameters, in other words, the single input corresponds to multiple outputs, it makes the neural network unable to find the optimal solution.

In order to overcome the problem of iterative convergence caused by the non-uniqueness of data, a feedback neural network [21] is used to optimize the backward design model in this paper. The optimized neural network is shown in Figure 6. The grating structural parameters marked in blue in Figure 4 are taken as the middle layer, which connects the forward simulation model and the backward design model to form a series feedback neural network. The backward design model of this optimized neural network still takes the spectral response as an input. It outputs the grating structure parameters and grating shape, as well as a predicted spectral response of the designed grating, which is used to evaluate the loss function.

The loss function of the series feedback neural network is defined as the mean square error between the input spectral response and the output spectral response:(4)$$Loss\;function=MSE\left(T,T^{\prime }\right)=\frac{1}{2}\sum_{i=1}^{N}{\left(t_{i}-t_{i}^{\prime }\right)}^{2},$$

For the series feedback neural network model, the contrast standard of the design result is no longer the grating structure parameters. It is just needed to compare the predicted spectral response of the design grating with the input. Therefore, the value of the loss function is reduced and the problem of non-convergence is avoided, overcoming the non-uniqueness of data.

#### 2.3.2. Parameters of the Neural Network

In order to improve the accuracy and calculation speed of the model, the parameters of the neural network are optimized. The number of hidden layers, the number of neural nodes, and the batch size were tested, respectively, and the results are shown in the following tables.

As shown in Table 3, the neural network cannot converge when the number of hidden layers is 5. When the number of hidden layers was 3, the neural network had less computational cost and a large loss function, which was 5 times that when the number of hidden layers was 4. Therefore, the number of hidden layers of the series feedback neural network was finally set as 4 in this paper. 

The test results of different neural nodes are given in Table 4. Considering the computational cost and the loss function comprehensively, the number of nodes in the four hidden layers was set to 128, 512, 512, and 128, respectively. In the output layer, 61 nodes with equal step lengths in the visible range of 0.4–0.7 μm were selected to characterize the spectral response. 

Batch size is the number of samples data required for neural network training, which affects the performance and calculation speed of the model. The smaller the batch size is, the higher the training accuracy of the neural network is. However, the iterations and the computational cost increase significantly. When the batch size is too large, the iteration may fail to converge. When the batch size is too small, memory explosion may occur. Therefore, an appropriate batch size should be selected to optimize the neural network. Table 5 shows the test results of different batch sizes. In this paper, the optimum batch size was 64, selected reasonably to get a balanced consideration of the computational cost and training accuracy. 

The optimized series feedback neural network with the above parameters was evaluated by the loss function, which was calculated by Equation (4) and shown in Figure 7. The value of the loss function decreased as the iterations increased. When it iterated 1000 times, the loss function dropped to 0.024. This is to say, the loss rate of test set data of the series feedback neural network was less than 2.4%. The optimized neural network has a good structure with low computational cost and small loss function.

## 3. Advantages of the Series Feedback Neural Network

The series feedback neural network is contrasted with the Rsoft software and FDTD software based on the same input of the two-dimensional subwavelength gratings. The computational cost and the CPU memory requirement were compared, and the results are shown in Table 6.

As the above Table shows, the computational cost consumed by the series feedback neural network was 1.12 s, almost 4.5 and 0.1% times that of the Rsoft and FDTD software, respectively. Meanwhile, the CPU memory requirement of the neural network was 31.2%, which was 31.8 and 32.0% times that of the Rsoft and FDTD software, respectively. In comparison, for the complex two-dimensional subwavelength gratings structure, the computational cost and CPU memory requirement of the series feedback neural network was much lower than that of the Rsoft and FDTD software. The series feedback neural network had good forward simulation and backward design ability, and its calculation performance was better than that of conventional design simulation software, such as the Rsoft and FDTD software.

## 4. Validations

To determine the feasibility and accuracy of the two-dimensional subwavelength grating design model based on the series feedback neural network, the forward simulation performance and reverse design ability of the model were validated, respectively.

### 4.1. Validation of the Forward Simulation

#### 4.1.1. Comparison with the RCWA Numerical Results

In order to verify the performance of the series feedback neural network to simulate and predict the two-dimensional subwavelength gratings, the spectral response curve simulated by the neural network is compared with the numerical simulation results of the rigorous coupled-wave analysis (RCWA); the comparison is shown in Figure 8. From Figure 8, the spectral response curve simulated by the neural network basically coincided with the results of RCWA numerical simulation, with a high consistence for each peak point. It indicates that the trained series feedback neural network is fully equipped with the function of simulation software. 

#### 4.1.2. Comparison with the Experimental Results

To verify the forward simulation ability of the series feedback neural network for data outside the data set, the simulation results of the neural network were compared with the experimental data in the literature [22]. The results are shown in Figure 9. Taking the design parameters of the grating, described in the literature as the inputs, the spectral response curve was obtained through the neural network forward simulation. It is compared with the spectral response measured in the literature.

The correlation analysis between the spectral response of neural network forward simulation and the literature data was conducted, as shown in Figure 9. The Euclidean distance [20] is:(5)$$d={\left[\sum_{k=1}^{n}{\left(x_{1k}-x_{2k}\right)}^{2}\right]}^{\frac{1}{2}},$$

The correlation coefficient is:(6)$$r=\frac{1}{1+d},$$

From the above equation, the correlation coefficients between the network simulation results and the experimentally measured values [22] of the three spectral response curves are 0.678, 0.702, and 0.654, respectively. According to the evaluation indicator shown in Table 7, there is a strong correlation between the network simulation results and the experimental data.

In conclusion, the simulated spectral response curves are in good agreement with the experimentally measured results, with the basically consistent peak points, showing that the neural network has high reliability for forward simulation of data outside the data set. Therefore, the performance of the series feedback neural network forward simulating the two-dimensional subwavelength gratings is confirmed.

### 4.2. Validation of the Backward Design Simulation

#### Data within the Test Set

Compared with the forward simulation, the backward design is always a pain point in two-dimensional subwavelength gratings design. In order to verify the backward design performance of the series feedback neural network, data in the test set were randomly selected as the target sample, and the verification results are shown in Figure 10. Taking the spectral response curves of 6 target samples as the input of the series feedback neural network, the geometric structure parameters of the two-dimensional subwavelength grating were obtained and shown in the upper left corner of Figure 10a–f, respectively.

In order to verify the correctness of the reverse design, we input the reverse designed grating geometric structure into the Rsoft software, calculate the spectral curve, and compare it with the target curve.

The Comparison between the predicted spectral response and the target spectral is shown in Figure 10a–f, respectively. It indicates that the backward design result is in good agreement with the target spectral response curve.

We also noticed that the consistency of different structures was quite different between target and predicted spectra. The main reason is that the training database used at present is too little to cover all the structures.

In general, the desirable design of the two-dimensional subwavelength gratings is the ideal spectral response curve, which has a distinct crest only at the peak point, and the other positions tend to be flat. Therefore, we take the single peak ideal Gaussian curve as the input, intending to verify the design performance of the series feedback neural network.

The proposed input ideal spectral response curve is the Gaussian curve within 0.4–0.7 µm, which is given by:(7)$$f=0.8exp\left[-\frac{{\left(x-x_{0}\right)}^{2}}{2\sigma^{2}}\right]+0.05,$$
where the $x_{0}$ was set as 0.46, 0.56, 0.61 respectively, and σ is 0.005.

Then, the three spectral curves obtained through Equation (7) were input into the series feedback neural network. Through the reverse design of the neural network, the geometric structure parameters and shapes of the two-dimensional subwavelength gratings were obtained. The spectral response curves of those grating are calculated by Rsoft software and compared with the ideal Gaussian curve, as shown in Figure 11. The calculated spectral response curves fit the ideal curve very well. According to Equations (5) and (6), the correlation coefficients between them are 0.701, 0.609, and 0.753, respectively. The design results are a strong correlation to the ideal curves. That is, the series feedback neural network has a good ability for backward design.

### 4.3. Sensing Characteristics

The refractive index is one of the main indicators of the characteristics of the solution. The two-dimensional grating structure designed in this paper has a high sensitivity to the refractive index. Figure 12 shows the changing trend of reflectivity in the range of 400–500 nm when the refractive index of the surrounding environment is changed from 1 to 1.6. For the changing trend of the peak wavelength, the Gaussian fitting method is used to calculate the peak mass center. The calculated position of the peak wavelength changes from 0.458 to 0.472 μm, and the absolute change of the wavelength peak position is 14 nm. If the conventional commercial optical fiber spectrometer is used to test this reflection, its resolution of wavelength can reach 0.02 nm [23], then the corresponding refractive index resolution is 8.57 × 10^–3^. This can meet the identification of various solutions such as methanol, ethanol, propanol, phenethyl ether, with its corresponding refractive index being 1.3290, 1.3618, 1.3593, and 1.3538, respectively, in that its difference of refractive index is more than refractive index resolution. 

## 5. Conclusions

Based on deep learning, a design method of a two-dimensional subwavelength grating filter was proposed. A series feedback neural network was built, which can realize forward simulation and backward design. The effects of the number of hidden layers, the number of neural nodes, and the batch size were studied, and the optimized network model structure was obtained as (5,5) × 128 × 512 × 512 ×128 × 61 and the batch size was 64. Through the training of 46,080 data sets generated by rigorous coupled wave analysis (RCWA) numerical simulation, the series feedback neural network can quickly and accurately simulate and design the two-dimensional gratings. The geometric structure parameters of the grating can be given within 1.2 s by the input response spectral curve. Through the calculation verification compared with the ideal Gaussian curve, numerical results and experimental results, the correlation coefficient is greater than 0.5, belonging to strong correlation. Moreover, the design efficiency of the series feedback neural network is greatly improved. The computational cost and CPU memory requirement decreased significantly, the former is only 4.5and 0.1% of the Rsoft and FDTD software, respectively, and the latter is about 31.8 and 32.0% of the two traditional software.

## Figures and Tables

**Figure 1 sensors-22-07758-f001:**
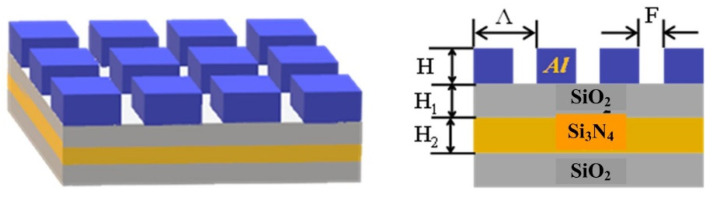
Structure diagram of the two-dimensional sub-wavelength grating.

**Figure 2 sensors-22-07758-f002:**
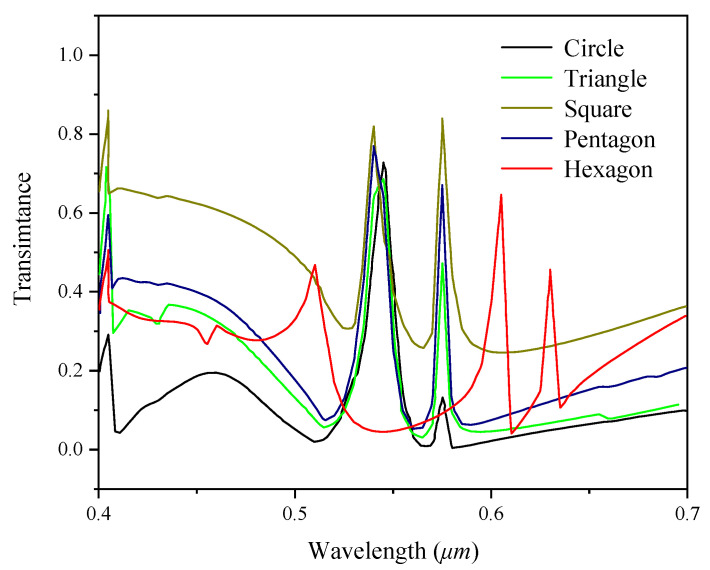
The influence of different grating shapes on spectral response (grating depth H = 0.04 µm, SiO_2_ waveguide layer thickness H_1_ = 0.08 µm, Si_3_N_4_ waveguide layer thickness H_2_ = 0.1 µm, duty ratio F = 0.65, grating period Λ = 0.25 µm).

**Figure 3 sensors-22-07758-f003:**
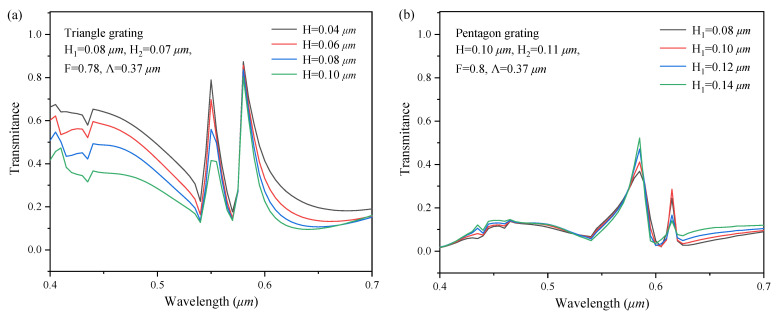
The spectral response curves of (**a**) triangle, (**b**) pentagon, (**c**) square with different H_2_, (**d**) square with different F, (**e**) square with different Λ, (**f**) different shapes with same parameters.

**Figure 4 sensors-22-07758-f004:**
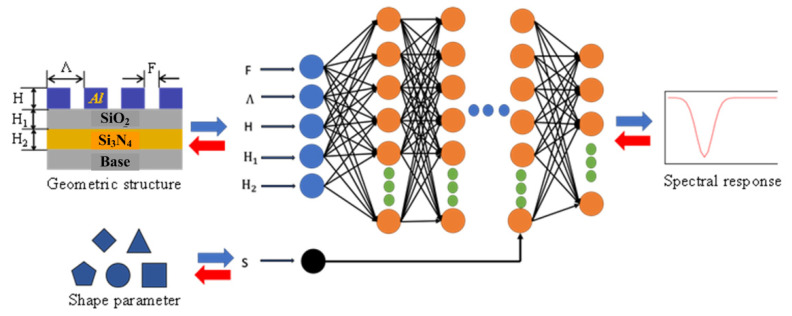
Schematic diagram of the two-dimensional sub-wavelength grating design neural network.

**Figure 5 sensors-22-07758-f005:**
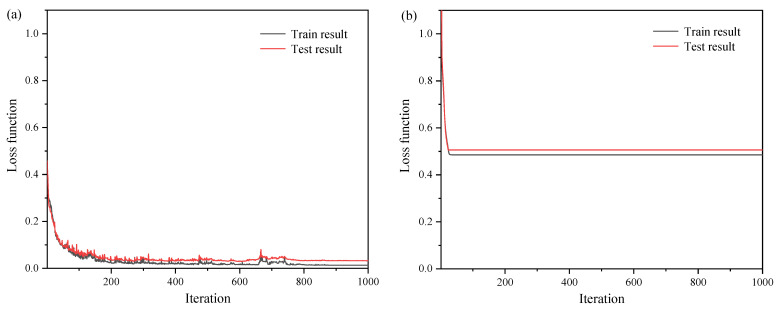
The loss function of the neural network: (**a**) the forward simulation model; (**b**) the backward design model.

**Figure 6 sensors-22-07758-f006:**
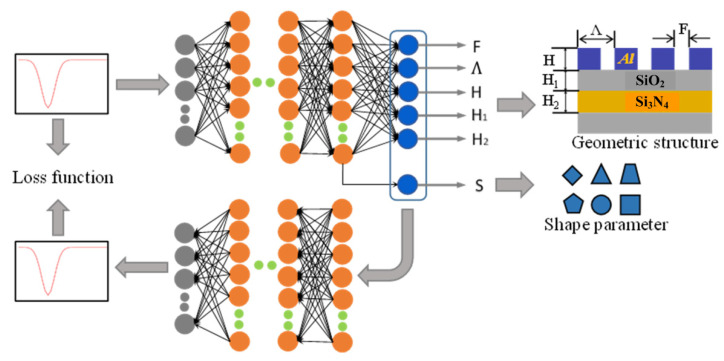
Schematic diagram of the series feedback neural network.

**Figure 7 sensors-22-07758-f007:**
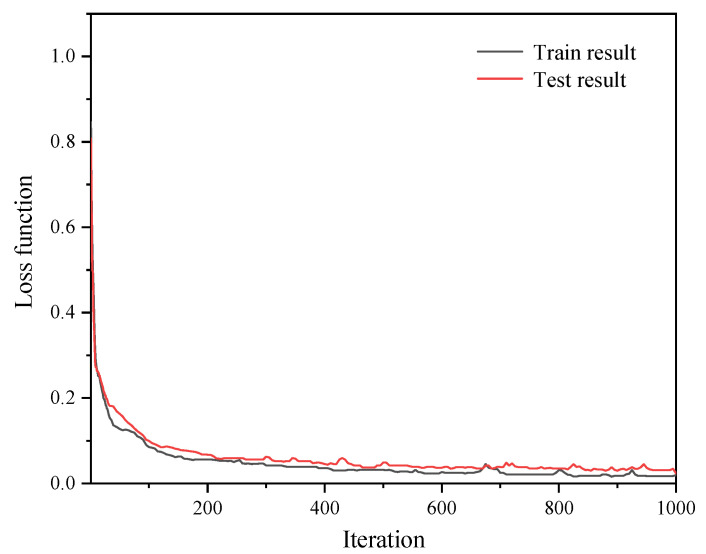
Loss function of the optimized series feedback neural network.

**Figure 8 sensors-22-07758-f008:**
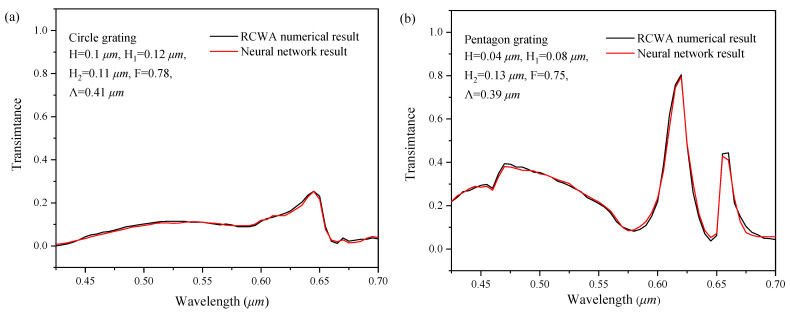
Comparison between the forward simulation and the RCWA numerical calculation for (**a**) circle gating with H = 0.06 µm, H_1_ = 0.12 µm, H_2_ = 0.11 µm, F = 0.78, Λ = 0.41 µm, (**b**) pentagon grating with H = 0.04 µm, H_1_ = 0.08 µm, H_2_ = 0.13 µm, F = 0.75, Λ = 0.39 µm, (**c**) square grating H = 0.06 µm, H_1_ = 0.08 µm, H_2_ = 0.09 µm, F = 0.78, Λ = 0.37 µm, (**d**) circle gating with H = 0.1 µm, H_1_ = 0.12 µm, H_2_ = 0.09 µm, F = 0.78, Λ = 0.37 µm, (**e**) circle gating with H = 0.04 µm, H_1_ = 0.12 µm, H_2_ = 0.11 µm, F = 0.67, Λ = 0.29 µm, (**f**) circle gating with H = 0.06 µm, H_1_ = 0.08 µm, H_2_ = 0.09 µm, F = 0.67, Λ = 0.27 µm.

**Figure 9 sensors-22-07758-f009:**
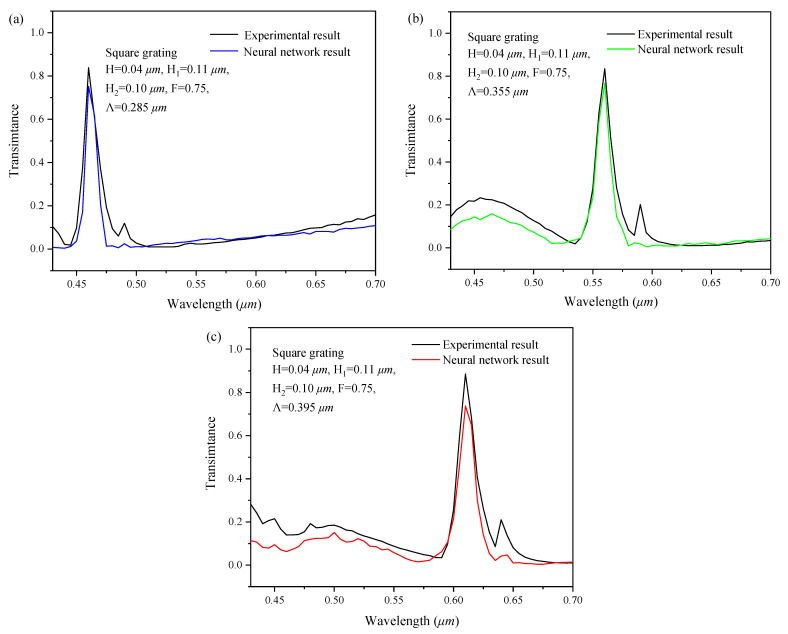
Comparison of the forward simulation with the experimental results in the literature [22] (peak wavelength: (**a**) 0.46 µm (**b**) 0.56 µm (**c**) 0.61 µm).

**Figure 10 sensors-22-07758-f010:**
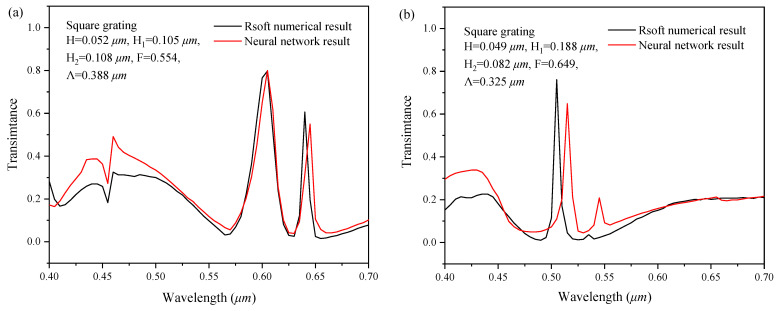
Validation of the inverse design, where (**a**–**f**) respectively represent the results of samples 1~6. The insert picture in the upper-left corner of each figure shows the designed structure parameters by the inverse network.

**Figure 11 sensors-22-07758-f011:**
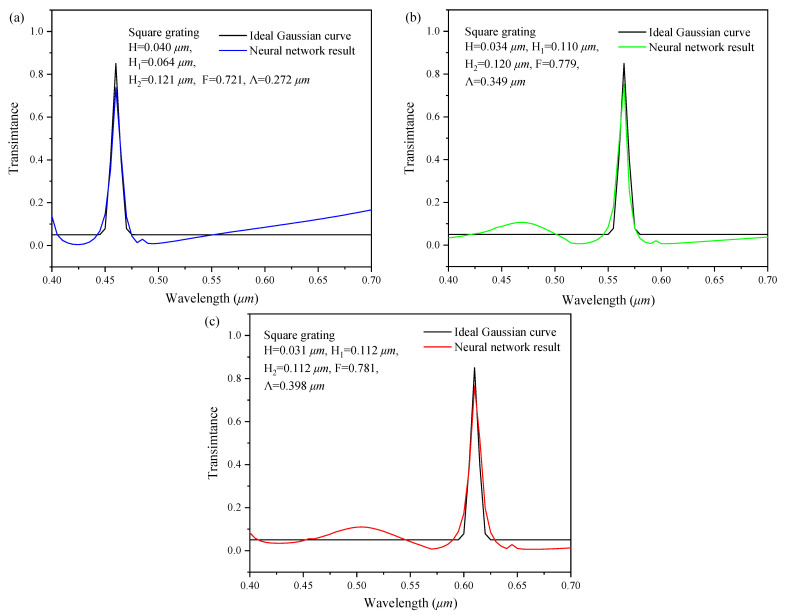
Validation of the ideal Gaussian curve. The peak wavelength: (**a**) 0.46 µm; (**b**) 0.56 µm; and (**c**) 0.61 µm. Insert picture in the upper-left corner of each figure shows the designed structure parameters by the inverse network.

**Figure 12 sensors-22-07758-f012:**
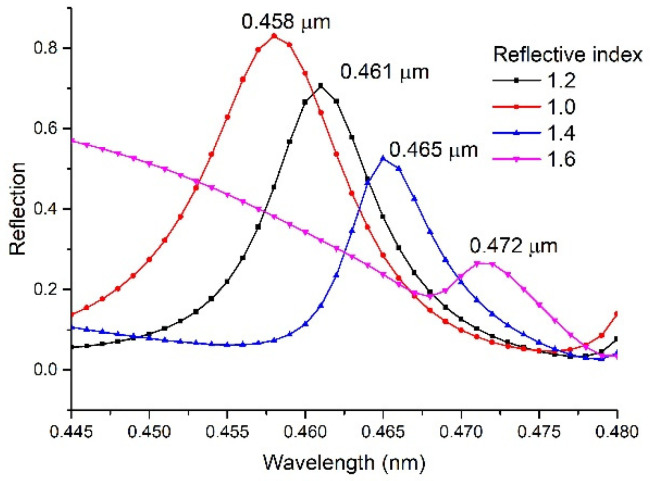
The trend of peak wavelength position with the refractive index.

**Table 1 sensors-22-07758-t001:** Geometric structure and shape parameters.

Parameters	Range
grating height, H	0.01–0.09 µm
SiO_2_ waveguide layer thickness, H_1_	0.09–0.19 µm
Si_3_N_4_ waveguide layer thickness, H_2_	0.05–0.15 µm
duty ratio, F	0.5–0.8
grating period, Λ	0.2–0.4 µm
grating shapes	circle, triangle, square, pentagon, hexagon

**Table 2 sensors-22-07758-t002:** The one-hot encoding of the grating shape.

Shape	Circle	Square	Triangle	Pentagon	Hexagon
digitizing	0	1	2	3	4
encoding	00001	00010	00100	01000	10000

**Table 3 sensors-22-07758-t003:** Test results of different hidden layers.

	Number of Hidden Layers	Computational Cost (s)	Loss Function
model 1	5	non-convergence	0.246
model 2	4	465.325	0.024
model 3	3	382.213	0.102
model 4	2	297.178	0.078

**Table 4 sensors-22-07758-t004:** Test results of different neural nodes.

	Network Structure	Computational Cost (s)	Loss Function
model 1	128 × 256 × 256 × 128	353.149	0.075
model 2	128 × 256 × 512 × 128	687.622	0.034
model 3	128 × 512 × 512 × 128	1035.211	0.024
model 4	128 × 512 × 1024 × 128	1335.112	0.033
model 5	128 × 1024 × 1024 × 128	1612.258	0.026

**Table 5 sensors-22-07758-t005:** Test results of different batch sizes.

	Batch Size	Computational Cost (s)	Loss Function
model 1	16	non-convergence	0.250
model 2	32	2368.581	0.044
model 3	64	1035.211	0.024
model 4	128	701.651	0.027
model 5	256	602.406	0.034

**Table 6 sensors-22-07758-t006:** Performance comparison of the series feedback neural network ^1^.

Type	Computational Cost (s)	CPU Memory Requirement (%)
neural network	1.12	31.2
Rsoft software	24.70	98.2
FDTD software	916.00	97.4

^1^ Based on Intel(R) Core (TM) I5−7400 CPU @3.00 GHz.

**Table 7 sensors-22-07758-t007:** Correlation coefficient evaluation indicator.

Range of the r Value	Correlation Level
0–0.1	non-correlation
0.1–0.3	weak correlation
0.3–0.5	moderate Correlation
0.5–1	strong correlation

## Data Availability

Not applicable.

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
