# Peer review of "A Design Method of Two-Dimensional Subwavelength Grating Filter Based on Deep Learning Series Feedback Neural Network"

_sensors, 2022, doi:10.3390/s22207758_

Round 1

Reviewer 1 Report

Dear author:

        I have carefully reviewed the manuscript of this article, and here are my comments on the manuscript.

Summary:

       In this paper, a new design method for two-dimensional subwavelength grating filters is proposed based on a series of feedback neural networks, which are learned and trained on the data set obtained from RCWA numerical simulations.

This design method avoids the step of solving Maxwell's equations in the traditional way, and is faster than the traditional method, but slightly less accurate. In addition, it enables both forward simulation and inverse design.

 Major criticisms:

Overall, the structure of this paper is clear, but in the fourth part, the advantages of the tandem feedback neural network should be placed at the end of the second part, which would make the structure of the paper more concise and standardized. The article contains an introduction, methodology, analysis, validation and conclusion, each part clearly illustrates the author's ideas, problem-solving approach and analysis process. The author answers the questions he/she posed in the previous section, clearly explains the methodology, and the paper is easy to understand.

The forward simulation results of the design method are highly correlated with the numerical simulation results, and the authors also characterize the correlation, but the correlation representation of the inverse design simulation results with the test set results in Section 3.2.1 is missing. As can be seen from Figure 10, most of the neural network simulation results deviate significantly from the test results, but Figure 10(d) shows a high degree of conformity, and the author does not have its analytical reasons.

Minor criticisms:

In section 3.2, the curve legend in the figure is a bit confusing. In the inverse design section, the result of the neural network should be the geometric parameters of the grating rather than the transmission curve, so the legend description should be slightly modified to make the picture easier to understand.

Author Response

Thank you very much for your review on our work. A great honor to reply to your comments. Please download the reply file in the attachment.

Reviewer 2 Report

In the considered article the authors consider a subwavelength grating filter structure to be optimized with a deep-learning series feedback neural network optimization method. Such methods are useful for optimizing structures containing many various parameters that are dependent on each other for the output and requiring high computational times. The authors consider forward, backward, and finally combination of the previous two to optimize their considered structure, which is an improvement of the previous two and named as series feedback neural network method. For validation the latter was compared with simulations and literature obtained experimental data obtaining a strong correlation. Further for impact importance compared to the simulation time compared to the common FDTD was reduced from 916s to 1.12s.

Overall, the article could be accepted. Due the fact that the optimal optical structures are getting more and more complex requiring us to think of newer methods to reduce simulation times. The neural network optimization methods that the authors considered is a rather good choice. The downside could be the lack of optimization for sensitivity or having minimal consideration of the considered structure as a sensor. Considering the journal name “sensors" the authors should address this. Just the initial paragraph seems lacking. If the authors believe it is not necessary they should comment why.

Points in the current manuscript which should be improved:

1. If possible, please improve the figures that contain graphs: quality of the text not always equal, ticks all inside or outside, only bottom x-axis and right-side y-axis to have ticks, too many legends in Fig.3(d)&(e).

2. Instead of Rsoft software the authors should method the simulation method that the Rsoft software used, for example Effective Index Method (EIM), and whether FDTD method was also as a simulation method package from Rsoft or that the authors built their own FDTD method.

3. There should be a spacing between numbers and SI units (ex. 0.07μm -> 0.07 μm).

4. Check for misspellings (ex. Monte-garlo -> Monte Carlo on line 69).

5. In Fig.1 cross-sectional view include the blue colored material name.

6. If the equations are taken from somewhere please refer them.

7. For the sensors journal the authors should include optimization for sensitivity or at least a guide how to utilize the considered method for sensitivity optimization.

Questions:

8.  Are there any other neural optimization methods?

9. Can the following method be used also for other passive optical structures? (Not filters; for example, grating couplers, arrayed waveguide gratings, mach-zhender interferometers, etc.)

10. Which mode was used for the simulations, is it fundamental TE, TM or hybrid?

Author Response

Thank you very much for your review on our work. A great honor to reply to your comments. Please download the reply files in the attachment. 

Round 2

Reviewer 2 Report

Considering the submitted article content and the scope of "Sensors" journal special issue "Optical Biosensors for Healthcare Monitoring" the match is too shallow to be accepted. 

I would suggest the authors to re-submit the article as an general article in  the sensors journal or in one of its special issues where the scope and article content match better. Another option is to make the submission to a journal which relies less on the sensors topic and more general to the optics devices (i.e. optics communications).  

*2 issue found in the revised manuscript:

1) Fig. 3,8,9,10,11 take too much space. Please use both left and right side column as it was done for Fig. 5.

2) The related content which was supposed to answer my general comment cannot be found in the page 19 or overall of the article. - This paragraph is not necessary if the authors decide to submit to a another journal.

Suggestion how to utilize considered method for optimizing for sensitivity: Multiply set of calculations by 2x. Where the second set is calculated with surrounding cladding that has slightly higher refractive index. Doing so for the considered "(sensor) filter" device in its transmission spectra authors will observe peaks being shifted. Next, based on the considered peak shift and division by the increase of cladding refractive index the authors can find the bulk sensitivity. Then, the authors can let the method to iterate for this bulk sensitivity while also setting a limit that the observed peak shift still has a peak shape.

Author Response

1 I would suggest the authors to re-submit the article as an general article in  the sensors journal or in one of its special issues where the scope and article content match better. Another option is to make the submission to a journal which relies less on the sensors topic and more general to the optics devices (i.e. optics communications).  

Thanks for your suggestions.

In our revised paper, the Sensing characteristics for the refractive index were included to match sensors topic better.

*2 issue found in the revised manuscript:

1) Fig. 3,8,9,10,11 take too much space. Please use both left and right side column as it was done for Fig. 5.

There are a lot of data, and the current format is not easy to confuse.

2) The related content which was supposed to answer my general comment cannot be found in the page 19 or overall of the article. - This paragraph is not necessary if the authors decide to submit to a another journal.

In our revised paper, we use the optimized structure to change the surrounding refractive index to simulate the sensing characteristics. The results show that the obtained two-dimensional grating structure has high refractive index sensitivity and can complete the identification of common solutions such as methanol.

Round 3

Reviewer 2 Report

Due the authors addressing including sensing characteristics I can agree that the submission could be accepted for the journal. 

The final say I would give however for the journal Editors. Given the manuscript having more impact for fast neural network optimization of filtering devices as can be read from the title.

If the manuscript is accepted and the authors wish to use the same optimization tool for further publication in the same journal I strongly suggest to modify the optimization tool to include also sensitivity related FOM. The obtained sensitivity (14nm/0.6RIU ~ 23nm/RIU) is actually a quite low value but as was mentioned previously by the authors. Having a large dataset of "optimized" filters we can pick the better ones where we have higher sensitivity. 

Without adding the reference the parameter which might be the FOM for this sensitivity value is optical space confinement factor (OSCF). This value can be obtained from a single simulation. It is the mode optical field ratio of outside of the device over the overall mode field. The higher OSCF the higher sensitivity since the more field interacts with sensing medium the higher sensitivity the filtering device will have.